# A conserved switch controls virulence, sporulation, and motility in *C. difficile*

**Michael A. DiCandia[1☉], Adrianne N. Edwards[1☉], Ysabella B. Alcaraz[1], Marcos P. Monteiro[1], Cheyenne D. Lee[1], Germán Vargas Cuebas[1], Pritha Bagchi[2], Shonna M. McBride[1]***

**1** Department of Microbiology and Immunology, Emory University School of Medicine, Emory Antibiotic Resistance Center, Atlanta, Georgia, United States of America, **2** Emory Integrated Proteomics Core, Emory University, Atlanta, Georgia, United States of America

☉ These authors contributed equally to this work.
* shonna.mcbride@emory.edu

**Data Availability Statement:** Sequence files were deposited to the NCBI Sequence Read Archive (SRA) BioProject PRJNA896704 under accession numbers SRX18115370, SRX18115371 and

## Abstract

Spore formation is required for environmental survival and transmission of the human enteropathogenic *Clostridioides difficile*. In all bacterial spore formers, sporulation is regulated through activation of the master response regulator, Spo0A. However, the factors and mechanisms that directly regulate *C. difficile* Spo0A activity are not defined. In the well-studied *Bacillus* species, Spo0A is directly inactivated by Spo0E, a small phosphatase. To understand Spo0E function in *C. difficile*, we created a null mutation of the *spo0E* ortholog and assessed sporulation and physiology. The *spo0E* mutant produced significantly more spores, demonstrating Spo0E represses *C. difficile* sporulation. Unexpectedly, the *spo0E* mutant also exhibited increased motility and toxin production, and enhanced virulence in animal infections. We uncovered that Spo0E interacts with both Spo0A and the toxin and motility regulator, RstA. Direct interactions between Spo0A, Spo0E, and RstA constitute a previously unknown molecular switch that coordinates sporulation with motility and toxin production. Reinvestigation of Spo0E function in *B. subtilis* revealed that Spo0E induced motility, demonstrating Spo0E regulation of motility and sporulation among divergent species. Further, 3D structural analyses of Spo0E revealed specific and exclusive interactions between Spo0E and binding partners in *C. difficile* and *B. subtilis* that provide insight into the conservation of this regulatory mechanism among different species.

## Author summary

*Clostridioides difficile* causes severe diarrheal disease and death in humans and livestock animals, and is a major public health concern. As an anaerobe, *C. difficile* transmission relies on the formation of hardy spores, while its pathogenesis requires the productions of toxins. Herein, we describe a previously unknown regulatory mechanism involving the sporulation factor Spo0E that controls spore and toxin production, as well as motility. We demonstrate that this multi-functional regulatory mechanism is also operational in the distant relative, *Bacillus subtilis*. The identification of specific interactions between Spo0E and the regulators RstA and Spo0A show how these factors work together to control

SAMN32933800. All other data are available in the manuscript or supplementary materials.

**Funding:** This research was supported by the U.S. National Institutes of Health (www.nih.gov) through research grants AI116933 and AI156052 to S.M.M., AI106699 to C.D.L. and G.V.C., DK126467 to G.V.C., GM008490 to M.A.D., GM149422 to Y.B.A., and AI179158 to M.P.M. The funders had no role in study design, data collection and analysis, decision to publish, or preparation of the manuscript. The content of this manuscript is solely the responsibility of the authors and does not necessarily reflect the official views of the National Institutes of Health.

**Competing interests:** The authors have declared that no competing interests exist.

toxin, motility, and spore formation. Moreover, the conservation of this system in other bacteria suggests that similar regulatory mechanisms exist in a wide range of species.

## Introduction

*Clostridioides difficile* is an anaerobic gastrointestinal pathogen that requires spore formation for transmission [1]. While spores are highly resistant to environmental insults, the formation of endospores is energetically costly and can result in long-term dormancy of the bacterium. Consequently, the initiation of spore development has evolved regulatory controls that prevent unnecessary dormancy. Though the regulatory pathways that control sporulation initiation in *Bacillus* species are well studied, the factors required for regulation of initiation in anaerobes, like *C. difficile*, are poorly conserved and remain incompletely defined [2].

One factor that is highly conserved and required for sporulation initiation in all spore formers is the transcriptional regulator, Spo0A [3]. In *Bacillus* species, Spo0A is directly inactivated by a small phosphatase known as Spo0E, which results in repression of spore formation [4]. However, Spo0E function has not been studied in the Clostridia or any other anaerobes, and as these systems regulate Spo0A through divergent mechanisms, the function of Spo0E in these organisms cannot be assumed [2,3].

In this work, we investigated the role of a predicted *C. difficile* Spo0E ortholog, CD630_32710 (CD3271), to determine its effect on sporulation initiation. Analysis of a *spo0E* mutant revealed that Spo0E represses sporulation of *C. difficile*, as was observed in *Bacillus*. Unexpectedly, we also observed that *C. difficile* Spo0E repressed motility and toxin production. Further investigation of Spo0E function revealed that Spo0E interacted directly with Spo0A, as predicted, but also interacted with the RRNPP (Rap-Rgg-NprX-PlcR-PrgX) regulator RstA. RstA was previously shown to directly decrease motility and toxin production as a transcriptional repressor and to induce sporulation through an undetermined mechanism [5,6]. These results reveal that Spo0E acts as a lynchpin in a mechanism that governs sporulation through interaction with Spo0A and concomitantly regulates toxin production and motility through its interaction with RstA.

Additionally, we determined that Spo0E promotes motility in *Bacillus subtilis*, indicating that Spo0E functions as a regulator of sporulation and motility in both species. Protein interaction models of *C. difficile* and *B. subtilis* Spo0E revealed specific interfacing residues between Spo0E and interacting partners that validate the functional studies of these regulators. Moreover, a search for Spo0E orthologs revealed widespread distribution among Gram-positive and Gram-negative bacteria. Together, these results suggest that Spo0E-like proteins are conserved among prokaryotes and represent an overlooked regulatory mechanism in bacteria.

## Results

### Spo0E represses sporulation, toxin production, and motility in *C. difficile*

To determine if the Spo0E ortholog has a role in *C. difficile* sporulation, we disrupted the predicted *spo0E* gene (S1 Fig and S1 Table) and assessed spore production in the mutant. The *spo0E* mutant sporulated at about twice the frequency of the wild-type (WT) parent strain, indicating Spo0E substantially represses sporulation in *C. difficile*, similar to *B. subtilis* (Fig 1A and 1B) [7].

Unexpectedly, it was also observed that colonies of the *spo0E* mutant appeared mucoid and spreading on agar plates, which was not reported previously for *spo0E* mutants in *Bacillus*

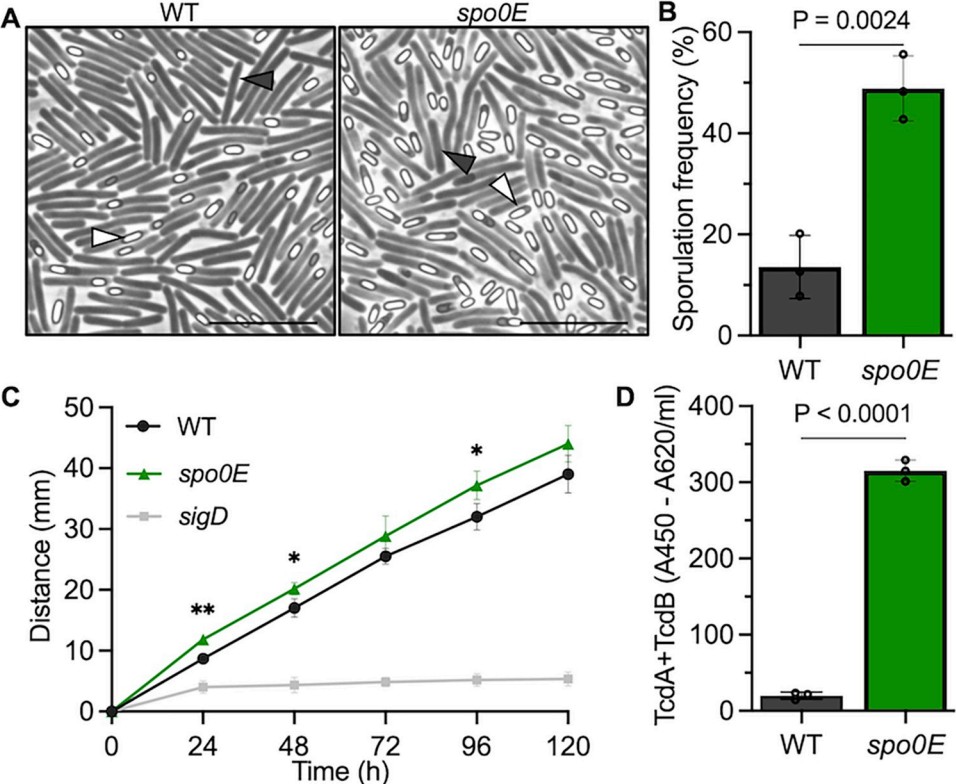

**Fig 1. Spo0E represses sporulation, motility, and toxin production in *C. difficile*.** A) Representative phase-contrast microscopy and B) sporulation frequencies of strain 630Δ*erm* (WT) and *spo0E* mutant (MC1615), grown on sporulation agar for 24 h. White triangles indicate phase bright spores, and dark triangles indicate vegetative cells. Scale bar: 10 μm. C) Swimming motility of 630Δ*erm* (WT), the *spo0E* mutant (MC1615), and the non-motile *sigD* mutant (RT1075; negative control). Active cultures were injected into soft agar and swim diameters measured daily for five days. D) Quantification of TcdA and TcdB from supernatants of 630Δ*erm* (WT) and the *spo0E* mutant (MC1615) grown in TY for 24 h. The means and SD of at least three independent experiments are shown. Unpaired *t*-tests were performed for B-D; *$P$ = <0.05, **$P$ = <0.01.

species [7–10]. The *spo0E* mutant colony phenotypes suggested that Spo0E could impact additional cellular processes. To explore this further, motility assays were performed on soft agar to assess the dissemination of the *spo0E* mutant over time, relative to the WT. As the spreading *spo0E* colony phenotype hinted, the *spo0E* mutant demonstrated increased motility on soft agar (Fig 1C), implicating *C. difficile* Spo0E in the regulation of motility. The *spo0E* phenotypes were fully complemented with the reintroduction of wild-type *spo0E* (Fig 2).

The primary driver of motility in *C. difficile* is the sigma factor, SigD, which also promotes expression of the genes encoding toxins TcdA and TcdB by driving transcription of the gene encoding the toxin-specific sigma factor, TcdR [11,12]. Considering the direct link between motility and toxin regulation, we next examined toxin production in the *spo0E* mutant using a TcdA/TcdB ELISA assay. The *spo0E* mutant produced markedly greater toxin (~15-fold) than the parent strain (Fig 1D). The increases in toxin and motility observed for the *spo0E* mutant strongly suggest that Spo0E represses SigD activity. The only factor that Spo0E-like proteins are known to interact with is the sporulation regulator Spo0A. However, such dramatic increases in toxin or motility are not observed for *spo0A* mutants [13], indicating that the effects of Spo0E on SigD-dependent regulation are independent of Spo0A, and thus, occur through an undescribed mechanism.

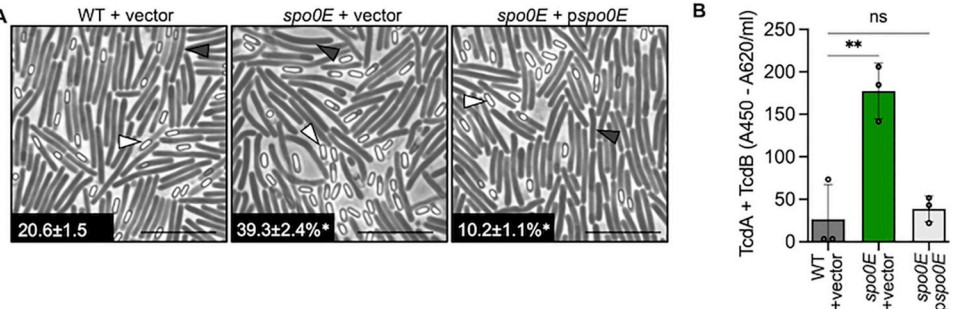

**Fig 2. *spo0E* phenotypes are complemented with reintroduction of *spo0E*. A)** Representative phase-contrast microscopy and sporulation frequencies of strain 630Δ*erm* + pMC123 (MC324), *spo0E* + pMC123 (MC1699), and complemented mutant *spo0E* + p*spo0E* (MC1698) grown on sporulation agar supplemented with 2 μg ml⁻¹ thiamphenicol for 24 h. n = 4 White triangles indicate phase bright spores, and dark triangles indicate vegetative cells. Scale bar: 10 μm **B)** Quantification of TcdA and TcdB from supernatants of strain 630Δ*erm* + pMC123 (MC324), *spo0E* + pMC123 (MC1699), and complemented mutant *spo0E* + p*spo0E* (MC1698) in TY supplemented with 2 μg ml⁻¹ thiamphenicol for 24 h. The means and SD of at least three independent experiments are shown and one-way ANOVA with Dunnett's multiple comparisons test was performed for B-D; $*P = <0.05$, $**P = <0.01$.

## Disruption of *spo0E* increases early production of toxins and morbidity during infection

The toxins TcdA and TcdB are responsible for *C. difficile* pathogenesis; thus, an increase in toxin synthesis within the host is expected to increase virulence. To determine if the *spo0E* mutant impacts virulence, a Syrian golden hamster model of *C. difficile* infection (CDI) was used to examine colonization, toxin production, and overall pathogenesis. Hamsters were infected with spores of 630Δ*erm* (WT) or the *spo0E* mutant, as described in the Methods, and monitored for symptoms of disease. Hamsters infected with *spo0E* mutant spores succumbed to infection faster than WT-infected animals (Fig 3A; median time to morbidity: 46.7 h for WT, 36.8 h for *spo0E*). To assess toxin production in the infected animals, fecal samples were collected 24 h post-infection and assayed for toxin content (Fig 3C), which revealed that the *spo0E* mutant generated significantly higher toxin loads within the intestine than WT early in infection. However, an analysis of toxin levels from moribund animals (Fig 3D) showed no overall increase in the toxin present between the *spo0E* mutant and parent strain, suggesting that the maximum threshold of toxicity is reached earlier in animals infected with the *spo0E* mutant. Further, examination of the *C. difficile* burden in moribund animals demonstrated that the number of *spo0E* mutant bacteria present in the cecum was less than the WT strain, but did not achieve statistical significance (Fig 3B). Thus, the increase in toxin production by the mutant was not due to greater colonization or carriage. Together, these data corroborate the *in vitro* toxin results and indicate that the *spo0E* mutant produces more toxin per bacterium *in vivo*, leading to more rapid morbidity.

## Spo0E interacts with regulators of sporulation, toxin, and motility

As mentioned, the virulence and motility phenotypes observed for the *C. difficile spo0E* mutant were not reported in prior studies of *spo0E* mutants in *Bacillus* species, which are only known to interact with Spo0A in the regulation of sporulation [4,8–10,14,15]. Because Spo0E had not been investigated in *C. difficile* or related anaerobes, no information is known about potential Spo0E-interacting partners that would facilitate the motility or toxin phenotypes. To this end, we sought to determine the proteins in the Spo0E interactome. Using a functional FLAG-tagged Spo0E expressed in the *spo0E* mutant, we performed co-immunoprecipitation (co-IP)

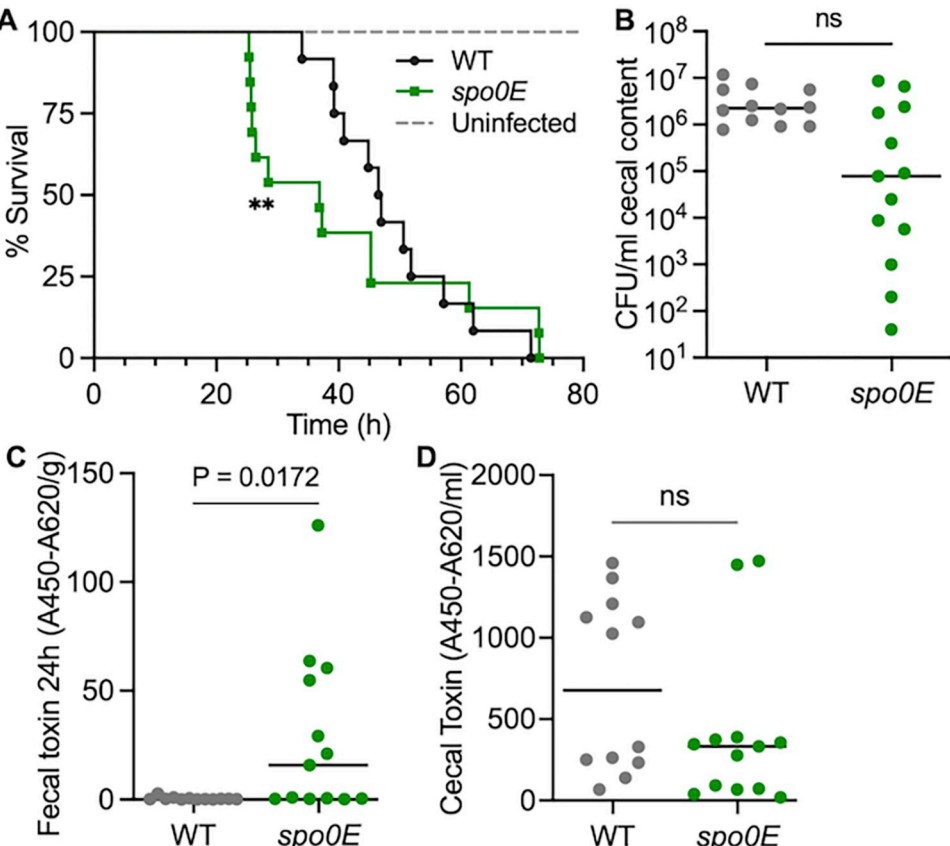

**Fig 3. Disruption of *spo0E* increases morbidity and early production of toxins during infection.** A) Kaplan-Meier survival curve representing the results from two independent experiments using Syrian golden hamsters inoculated with 5000 spores of *C. difficile* strain 630Δ*erm* (WT, n = 12) or *spo0E* mutant (MC1615, n = 13). Mean times to morbidity: WT, 48.7 ± 10.7 h; *spo0E*, 40.7 ± 17.8 h. **P < 0.001, Log-rank test. B) Total *C. difficile* CFU/ml of cecal content recovered post-mortem (ns = not significant; unpaired *t* test). ELISA quantification of TcdA and TcdB toxin per C) gram of fecal sample collected 24 h post-infection or D) per ml of cecal content collected post-mortem. Mid-line indicates median toxin values; unpaired *t*-test. Data for wild-type infected animals were previously published as part of the manuscript Infect Immun 91:e00319-23.

of Spo0E from *C. difficile* grown on sporulation agar to the onset of sporulation (12 h). Spo0E-FLAG was purified from cells and assessed for proteins bound to Spo0E by LC-MS/MS analysis. Few proteins were significantly enriched in the Spo0E pulldowns relative to negative controls, and only a handful of proteins were both enriched and abundant by LC-MS/MS counts (S2 Fig and S2 Table). As expected, the most abundant protein identified from the Spo0E pulldowns was the sporulation regulator, Spo0A, which suggests that *C. difficile* Spo0E directly regulates Spo0A activity as observed in *Bacillus* species (Table 1). But in addition to Spo0A, the regulator RstA was also a highly abundant Spo0E-interacting protein. RstA is a multifunctional, RRNPP-family protein that represses toxin and motility in *C. difficile* by directly controlling transcription of motility genes, *sigD*, *tcdR*, *tcdA*, and *tcdB* [5,6,16]. RstA also promotes sporulation through an independent regulatory domain [6], although the mechanism by which RstA functions to regulate sporulation was not understood. These results imply that the mechanism through which Spo0E controls toxin and motility is by its interaction with RstA and conversely, the regulation of sporulation by RstA is facilitated by its interaction with Spo0E. The translation factor EF-4 (LepA) was also highly enriched in the Spo0E pulldown (**Table 1**), though the significance of this interaction is not apparent.

**Table 1. Enriched factors bound to Spo0E or Spo0A in *C. difficile*[†].**

| co-IP Target | Bound Proteins | -Log P-value | Log$_2$ Intensity/ control |
|---|---|---|---|
| Spo0E-FLAG | Spo0A | 8.0 | 1.4 |
| | LepA | 7.0 | 1.3 |
| | Spo0E | 6.7 | 1.7 |
| | RstA | 5.5 | 1.3 |
| Spo0A-FLAG | PtpC | 5.5 | 1.5 |
| | Spo0A | 5.5 | 1.3 |
| | Spo0E | 5.3 | 1.5 |
| | CD630_12310 | 5.2 | 1.2 |

[†]Proteins identified through co-immunoprecipitation of Spo0E-FLAG or Spo0A-FLAG and LC-MS/MS analysis.

To accompany the Spo0E co-IP, we performed Spo0A-FLAG pulldowns from sporulating cells and found that Spo0E was similarly highly enriched in the LC-MS/MS results (Tables 1 and S3). However, RstA was not reliably enriched with Spo0A-FLAG, indicating that Spo0E can serve as an intermediate between RstA and Spo0A. In addition to the peptide analyses, we performed independent Spo0A and Spo0E pulldowns followed by western blot analyses for binding partners (Fig 4) These findings further support a model by which Spo0E influences sporulation, toxin, and motility through interactions with both Spo0A and RstA. In addition, the phosphotransfer protein PtpC co-purified with Spo0A, as previously observed *in vitro* [17], and CD630_12310, a predicted site-specific recombinase, was also highly enriched in the Spo0A pulldown.

The identified interactions of Spo0E with RstA and Spo0A strongly suggested that the interface between these factors is necessary for regulation of the motility, toxin, and sporulation phenotypes by Spo0E. To explore these interactions further, we employed AlphaFold modeling using the *C. difficile* Spo0E, Spo0A, and RstA proteins. Spo0E was predicted to interface simultaneously with the C-terminal tetratricopeptide repeat (TPR) domain of RstA and the

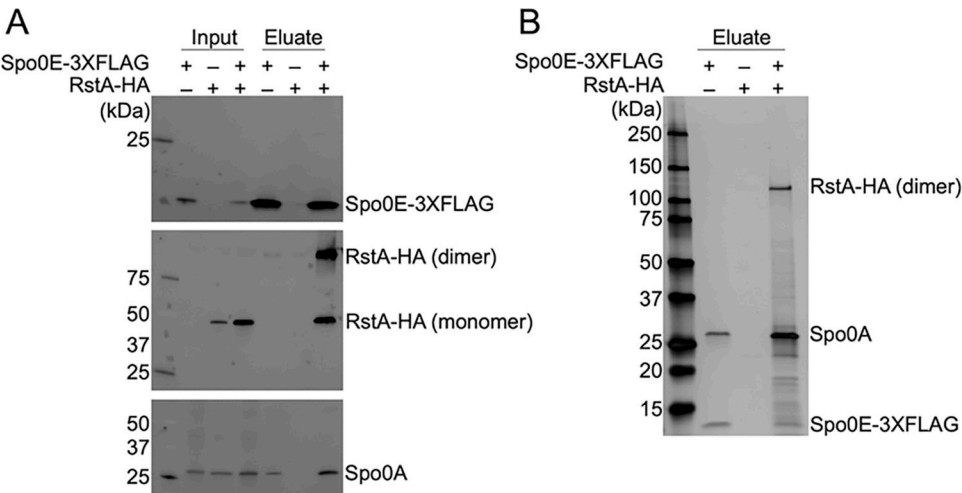

**Fig 4. Spo0E directly interacts with Spo0A and RstA.** Representative α-FLAG, α-HA, and α-Spo0A western blots of lysates (input) and eluates (A) or representative silver stain of eluates (B) obtained from 3XFLAG-specific co-immunoprecipitations of wild-type 630Δ*erm* strains expressing either *spo0E*-3XFLAG (MC2695), *rstA*-HA (MC2696), or both *spo0E*-3XFLAG and *rstA*-HA (MC2697) grown on 70:30 agar at H$_{12}$. At least three independent biological replicates were performed.

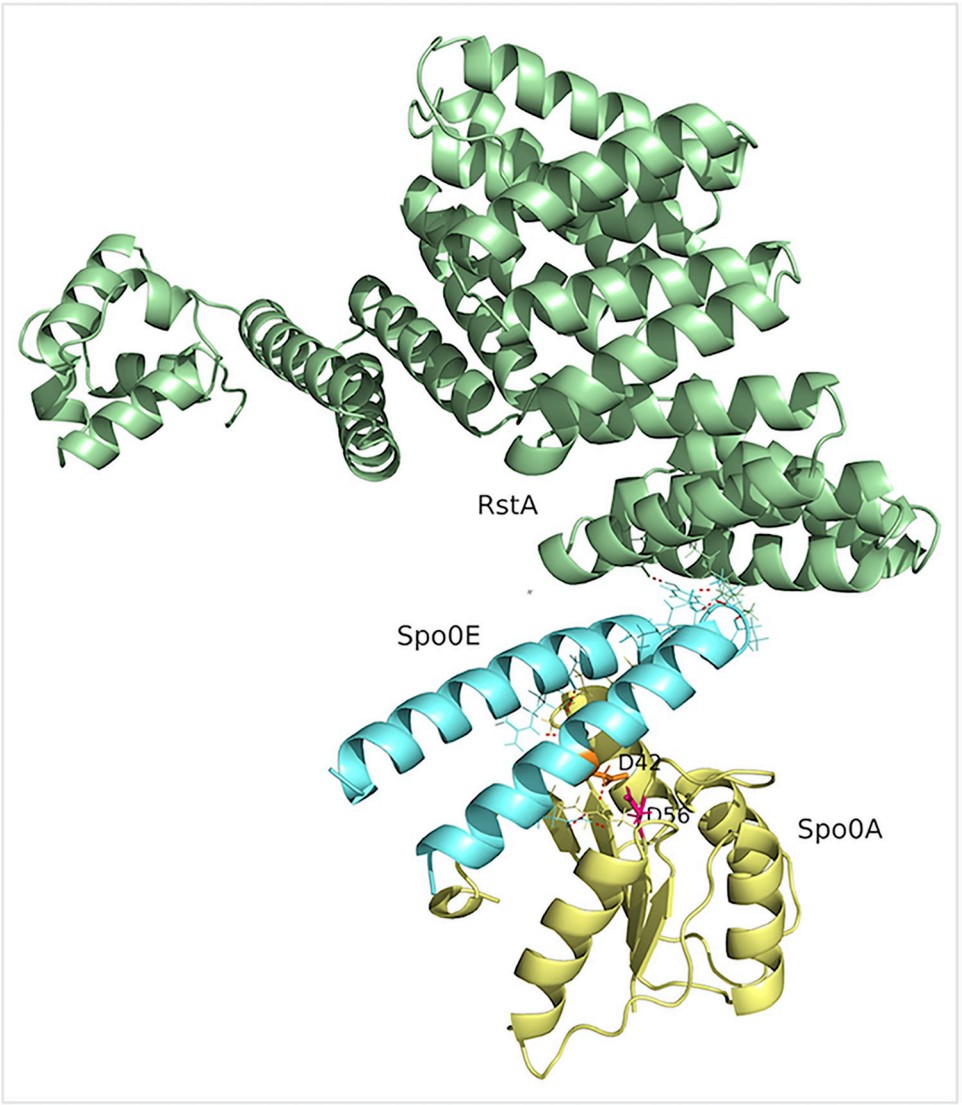

**Fig 5. *C. difficile* Spo0E is predicted to interact with functional domains of Spo0A and RstA.** Predicted structural interface of *C. difficile* Spo0E with Spo0A and RstA. The Spo0A N-terminal receiver domain (amino acids 5–115) is shown in yellow, with the conserved phosphorylation site (D56) labelled in pink. Spo0E is shown in blue with the conserved D42 of the SxxxD motif highlighted in orange. RstA is shown in green. Red dashes indicate polar contacts between amino acids. Spo0A Uniprot ID: Q18B74, Spo0E Uniprot ID: A0A7Y0LUW8, edited in PyMol (PyMOL Molecular Graphics System, Version 2.0 Schrödinger, LLC).

N-terminal receiver domain of Spo0A (Fig 5 and S4 Table). Both helices of Spo0E interface with Spo0A, including the conserved aspartate residue of Spo0E, D42, of the signature SxxxD motif, which is critical for Spo0E function in *B. subtilis* (S3 Fig) [4,8,9,15,18]. The interactions between Spo0E and Spo0A appear to block the Spo0A phosphorylation site (D56), which is consistent with the role of Spo0E as an inhibitor of Spo0A activation (*9, 19*). Most of the residues of Spo0A that interact with Spo0E have established sporulation phenotypes in prior mutagenesis studies [8,18,19]. The interactions of Spo0E with the C-terminal TPR domain of RstA occur within and near the linker region of the two Spo0E alpha-helices, and are distinct from interactions with Spo0A. The C-terminal region of RstA is predicted to respond to quorum sensing (QS) signals based on similarity with other RRNPP regulators; however, no

cognate QS signal has been identified [5,6,20]. No specific interactions were apparent between RstA and Spo0A from these or other experimental results.

## Spo0E regulation of sporulation and motility are facilitated through species-specific interactions

Considering the evidence that Spo0E interfaces with multiple regulatory factors to control different physiological processes in *C. difficile*, we questioned whether Spo0E has similar functions in other species that were not investigated in prior studies. For this, we revisited the original resource for Spo0E function, *B. subtilis*. *B. subtilis* is the model organism for endospore formation and is motile; however, it does not produce human pathogenic toxins. As *B. subtilis spo0E* mutants already have a verified hypersporulation phenotype [7,9], we assessed the mutant for motility. Using *B. subtilis* wild-type and an isogenic *spo0E* deletion mutant, we examined motility on soft agar plates for 24 h (Fig 6). The *B. subtilis spo0E* mutant consistently exhibited reduced swimming motility on soft agar. However, no decrease in motility was observed for the *spo0A* mutant, suggesting that the *spo0E* motility phenotype is also

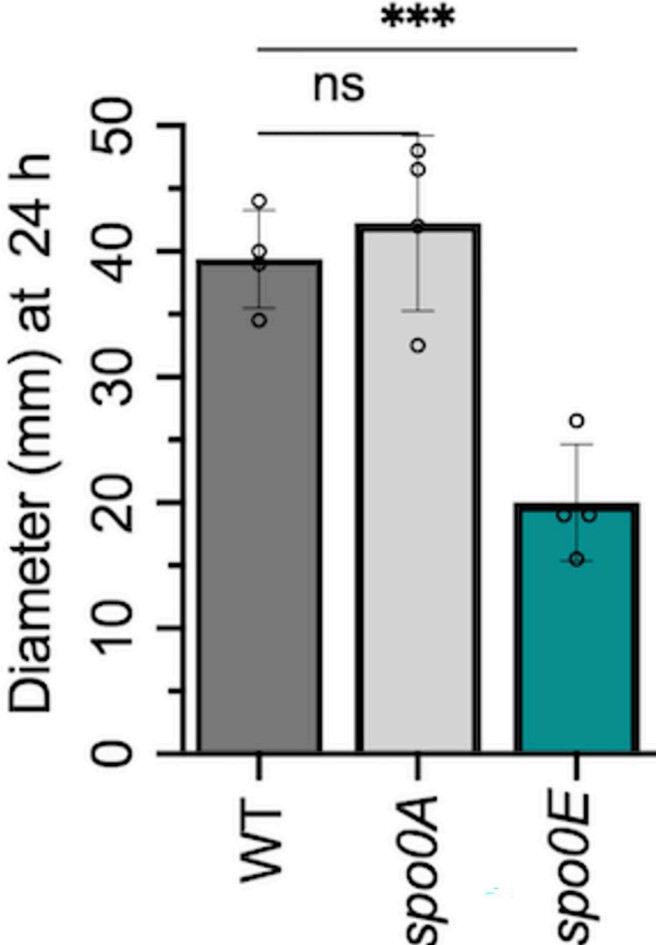

**Fig 6. Spo0E regulates motility in *B. subtilis*.** Swimming motility of *B. subtilis* IAI (WT), *spo0A* (MC2261), and *spo0E* mutant (MC2400). Active cultures were injected into soft agar and swim diameters measured after 24 h. The means and SD of at least three independent experiments are shown. A one-way ANOVA with Dunnett's multiple comparisons test was performed; $*P = <0.05$, $***P = <0.001$.

independent of *spo0A* in *B. subtilis*. The results suggest that *B. subtilis* Spo0E promotes motility, while *C. difficile* Spo0E suppresses motility (Fig 1C). Thus, Spo0E differentially regulates motility in multiple species, in addition to its role in the regulation of sporulation.

To better understand how *B. subtilis* and *C. difficile* Spo0E regulate motility and sporulation differently, cross-complementation studies were performed for the *spo0E* mutants of both species, followed by assessment of functions. No significant complementation of function was observed for either *B. subtilis* or *C. difficile* expressing the corresponding orthologous Spo0E (S4 Fig). The lack of cross-complementation suggested that interactions between Spo0E and the respective regulators have co-evolved for species-specific functionality. The receiver domains of *B. subtilis* and *C. difficile* Spo0A share 47% similarity, while the equivalent RRNPP regulator to RstA in *B. subtilis* is unknown or absent. Using AlphaFold modeling, we compared and contrasted the Spo0E-Spo0A interactions for *C. difficile* and *B. subtilis* (S5–S7 Tables), which revealed that key interfacing residues were absent for cross-species interactions of Spo0E and Spo0A for both species. These data highlight the species-specific residues and interactions that define Spo0E function in two divergent species.

## Spo0E-like proteins are conserved and prevalent across phylogenies

To understand the broader role of Spo0E, we searched for Spo0E orthologs in other species. The Spo0E family of proteins contain a signature five amino acid motif (SxxxD) [4,8,9,15,18]. To identify Spo0E orthologs, we probed for the Spo0E signature motif using AlphaFold and PSI-BLAST, and filtered by proteins that were between 40–100 amino acids in length (*C. difficile* Spo0E is 53 amino acids in length) (S5A Fig) [8,21–23]. We then predicted the 3D structure of proteins that met these criteria using Phyre2, comparing these to the known *Bacillus* Spo0E structure that is comprised of 85 aa and organized as two α-helices connected by a loop (S5B and S5C Fig) [15]. The presence of Spo0E orthologs encoded in the genomes of Gram-positive and Gram-negative bacteria with and without motility and sporulation abilities (S5A Fig) suggests that Spo0E-like proteins perform diverse regulatory functions that may be species specific.

## Discussion

In this study, we identified an ortholog to the *Bacillus* Spo0E protein and investigated its role in *C. difficile* physiology and pathogenesis. We established that *C. difficile* Spo0E represses sporulation, as was observed in *Bacillus* species [4,7–9]. In addition, we found that Spo0E represses *C. difficile* toxin production and motility, which was not recognized in prior Spo0E studies of *Bacillus*. By assessing the Spo0E interactome, we discovered direct interactions between Spo0E and Spo0A, as well as Spo0E and the regulator RstA. Identification of this interacting triad illuminates the molecular mechanism through which RstA promotes spore formation and Spo0E represses toxin production and motility in *C. difficile*. This mechanism supports a new model for regulatory coordination of motility, virulence, and sporulation in *C. difficile* (Fig 7), wherein Spo0E binds to Spo0A to inhibit premature sporulation and binds to RstA to repress motility and toxin production until cells transition to stationary phase. Although Spo0E appears to bind Spo0A and RstA at different residues, it is not clear how the interactions of these factors are controlled. It is also not apparent whether Spo0E binding to both Spo0A and RstA simultaneously is important for function. The sporulation, motility, and toxin phenotypes of the *C. difficile spo0E* mutant strongly suggest that RstA-Spo0E and Spo0A-Spo0E interactions change during stationary phase to allow for sporulation, toxin production, and motility. These phenotypic changes are likely brought on by the interaction of

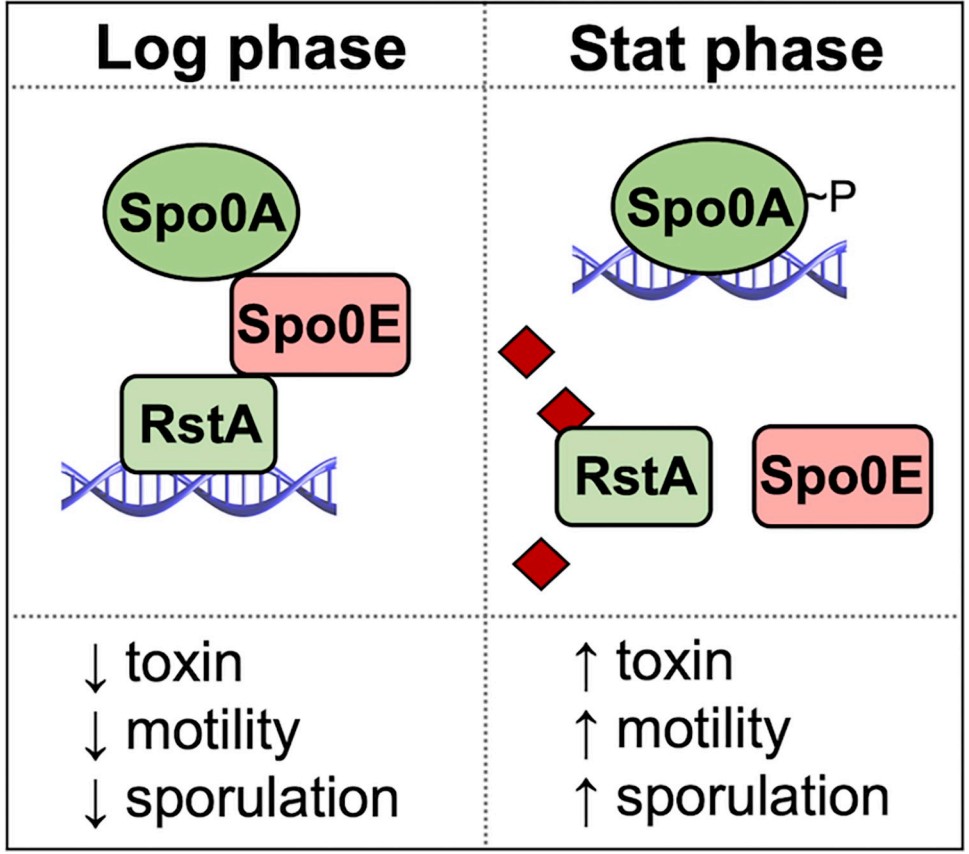

**Fig 7. Model of Spo0E influence on sporulation initiation in *C. difficile*.** During exponential growth, Spo0E binds Spo0A and RstA, preventing Spo0A activation of sporulation, and promoting RstA repression of toxin and motility genes (P*flgB*, P*tcdR*, P*tcdA*, P*tcdB*). At the transition to stationary phase, RstA is predicted to bind a quorum sensing peptide (♦), resulting in confirmational changes that prevent Spo0E protein-protein interactions. This leads to derepression of toxin and motility genes and the phosphorylation of Spo0A, resulting in sporulation.

RstA with a quorum sensing peptide, though experimental challenges have impeded identification of an RstA-binding signal.

The discovery of this mechanism introduces many questions about the regulatory role of Spo0E in other species. The identification of dual roles for Spo0E in *B. subtilis* motility and sporulation suggests broad conservation of Spo0E function as a regulator of these processes in endosporulating Firmicutes. The interaction of Spo0E with the RRNPP regulator, RstA, suggests that Spo0E orthologs may bind to other RRNPP-family proteins. RRNPP regulators control diverse physiological processes in bacteria, including toxin expression, nutrient acquisition, biofilm formation, solventogenesis, motility, sporulation, and competence in response to binding small quorum sensing peptides [20,24–28]. Many of the RRNPPs interact with response regulators or directly facilitate transcription of genes that direct the above processes (*e.g.*, Rap, Rgg, NprR, PrgX, PlcR) [20]. Spo0E ortholog interactions with response regulators or RRNPPs, or both, could add a layer of regulatory control that interfaces with other physiological processes, as Spo0E does in *C. difficile*. However, the specific interactions and interfaces between RRNPP proteins and response regulators are not well conserved, and given the divergence in Spo0E ortholog sequences, we expect similar diversity in the interactions between Spo0E and their partners in other species.

Through phylogenetic analyses, we identified Spo0E-like proteins in many Gram-positive and Gram-negative bacteria, as well as in the Archaea (S5 Fig). The presence of Spo0E in species that do not sporulate or are non-motile suggests the evolution of divergent functions for Spo0E in other systems. While the role of these Spo0E orthologs is not known, a plausible interaction in any of these systems would involve contact with a conserved partner protein, such as a response regulator. Given the scarcity of experimental data on Spo0E-like proteins and the diversity of regulatory functions already identified for Spo0E, future studies should consider a variety of possible regulatory mechanisms for the function of Spo0E orthologs.

# Materials and methods

## Ethics statement

All animal experimentation was performed under the guidance of veterinarians and trained animal technicians within the Emory University Division of Animal Resources (DAR). Animal experiments were performed with prior approval from the Emory University Institutional Animal Care and Use Committee (IACUC) under protocol PROTO201700396. Male and female Syrian golden hamsters (*Mesocricetus auratus;* 6–8 weeks old) were purchased from Charles River Laboratories and housed in sterile, individual cages in an animal biosafety level 2 facility. Animals considered moribund based on defined endpoints were euthanized in accordance with the Panel on Euthanasia of the American Veterinary Medical Association. The University is in compliance with state and federal Animal Welfare Acts, the standards and policies of the Public Health Service, including documents entitled "Guide for the Care and Use of Laboratory Animals" National Academy Press, 2011, "Public Health Service Policy on Humane Care and Use of Laboratory Animals" September 1986, and Public Law 89–544 with subsequent amendments. Emory University is registered with the United States Department of Agriculture (57-R-003) and has filed an Assurance of Compliance statement with the Office of Laboratory Animal Welfare of the National Institutes of Health (A3180-01).

## Bacterial strains and growth conditions

Bacterial plasmids and strains used in this study are listed in (S1 Table). *C. difficile* was routinely grown in BHIS or BHIS supplemented with 2–5 µg ml$^{-1}$ thiamphenicol or 5 µg ml$^{-1}$ erythromycin for selection, as needed (Sigma).[29] Active *C. difficile* cultures were supplemented with 0.1% taurocholate (Sigma) and 0.2% fructose to stimulate germination and prevent sporulation prior to assays.[29,30] *C. difficile* was grown on 70:30 agar for sporulation assays as previously described.[30] *C. difficile* was cultivated in a 37°C anaerobic chamber (Coy) with an atmosphere consisting of 10% H$_2$, 5% CO$_2$, and 85% N$_2$, as previously described.[31] *B. subtilis* strains were grown in LB or Difco sporulation medium (DSM) at 30–37°C, supplemented with 1 µg ml$^{-1}$ erythromycin, 7 µg ml$^{-1}$ kanamycin, or 100 µg ml$^{-1}$ spectinomycin, as needed. Strains of *Escherichia coli* were grown in LB at 30–37°C, supplemented with chloramphenicol 20 µg ml$^{-1}$, ampicillin 100 µg ml$^{-1}$, or spectinomycin 100 µg ml$^{-1}$ as needed.[32] Kanamycin 100 µg ml$^{-1}$ was used for *E. coli* HB101 pRK24 counterselection after conjugation with *C. difficile*.[33]

*C. difficile* 630 strain (GenBank accession AJP10906.1) was used as a template for primer design, and *C. difficile* 630Δ*erm* genomic DNA was used for PCR amplifications. S8 Table contains oligonucleotides used in this study. *C. difficile* mutants and complements were generated by conjugation, followed by selection and PCR confirmation as previously described.[34–36] The *C. difficile* spo0E mutant was created by Targetron insertion within the coding sequence as previously detailed.[37] The *B. subtilis* spo0E mutant was generated by natural competence

transformation of strain 1A1 with genomic DNA from strain BKE13640. Vector construction details are outlined in S9 Table.

## DNA extraction and hybrid sequencing analysis

Genomic DNA was extracted as previously described.[38] Library prep and sequencing for both Illumina and Oxford Nanopore Technologies (ONT) samples was performed by the Microbial Genomics Sequencing center (migscenter.com). Whole genome sequencing variant calling was performed using paired-end reads generated by Illumina sequencing (2 x 150 bp). Reads were trimmed using the BBDuk plug-in in Geneious Prime v2022.2.2, then mapped to the reference genomes NC_009089 (*C. difficile*) or NC_000964 (*B. subtilis*) and the respective parent strains (https://www.geneious.com).

The Bowtie2 plugin was used to search for the presence of SNPs or InDels in the *C. difficile* *spo0E* mutant under default settings with a minimum variant frequency set at 0.95, and no variants of concern were identified relative to the parental strain.[39] A *de novo* assembly of Illumina and ONT reads was then performed to confirm that the Targetron::*ermB* was inserted solely within the coding region of *spo0E*. Assembly was performed using Unicycler under default settings.[40] The assembled genome was annotated to the reference genome (NC_009089) using Geneious Prime. Circos plot was generated using PATRIC web resources.[41,42] Genome sequence files were deposited to the NCBI Sequence Read Archive (SRA)_BioProject PRJNA896704 under accession numbers SRX18115370, SRX18115371 and SAMN32933800.

## Sporulation assays

*C. difficile* ethanol-resistance sporulation assays were performed on 70:30 sporulation agar as previously described.[6,43,44] Briefly, assessed strains were grown in BHIS broth supplemented with 2 μg ml$^{-1}$ thiamphenicol as needed. Log-phase cultures were diluted with fresh BHIS, grown to an $OD_{600} = 0.5$, and plated onto 70:30 sporulation agar supplemented with 2 μg ml$^{-1}$ thiamphenicol, if needed for plasmid maintenance. After 24 h growth, cells were suspended in BHIS and total vegetative cells were enumerated by plating on BHIS agar. Concomitantly, 0.5 ml of resuspended cells were exposed to a mix of 0.3 ml 95% ethanol and 0.2 ml dH$_2$O for 15 min, serially diluted in 1X PBS and 0.1% taurocholate, and plated onto BHIS agar with 0.1% taurocholate to determine the total spores per ml. CFU were counted after 36 h of outgrowth, and sporulation frequency was calculated as the proportion of spores that germinated after ethanol treatment divided by the total number of cells.[6] A *spo0A* mutant was included as a negative control for all experiments to demonstrate vegetative cell death following ethanol treatment. Statistical significance was determined using a one-way ANOVA with Dunnett's multiple comparisons test in GraphPad Prism v9.0.

*B. subtilis* sporulation was assayed as previously described, with some modifications [45]. Strains were grown in LB broth to an $OD_{600}$ of 0.5 and 1 ml was used to inoculate 49 ml of DSM broth at 37˚C, 225 RPM for aeration. Sporulation cultures were assessed for growth until reaching an $OD_{600}$ of 1.0 ($T_0$ / stationary phase), followed by 24 h of growth with aeration. Cultures were then assessed for ethanol-resistant spores as outlined for *C. difficile*, and enumerated after outgrowth on LB agar. Due to lysis of vegetative cells prior to $T_8$, spore CFU/ml were calculated, rather than spores per total cells. Statistical significance was determined using a one-way ANOVA with Dunnett's multiple comparisons test in GraphPad Prism v9.0.

## Toxin quantification

Quantification of TcdA and TcdB toxins was assessed from *C. difficile* culture supernatants as previously described.[16] Briefly, cultures were grown for 24 h in TY media pH 7.4,

supplemented with 2 μg ml$^{-1}$ thiamphenicol as needed. Total toxin was assessed in technical duplicate using a *C. difficile* toxin ELISA kit (tgcBIOMICS) according to manufacturer's instructions. The technical duplicate measurements were averaged for a minimum of three biological replicates. A two-tailed Student's *t*-test was performed to determine statistical significance using GraphPad Prism v9.0. Toxin production was quantified from fecal and cecal samples of animals using the same assay with minor modifications. Feces collected from live animals or cecal contents recovered immediately post-mortem were stored at 4˚C prior to assay. Fecal samples were weighed to calculate toxin levels per gram of feces, then resuspended in 450 μl of Dilution Buffer. Cecal contents were diluted either 1:10 or 1:40 in Dilution Buffer, and toxin levels were normalized per ml of cecal content. A two-tailed Student's *t*-test was performed to determine statistical significance in toxin levels between both wildtype and *spo0E* fecal and cecal samples using GraphPad Prism v9.0.

## Motility assays

Swimming motility assays were performed as previously described with minor modification of inoculum size.[6] Cultures of *C. difficile* or *B. subtilis* were grown in BHIS or LB broth, respectively, to an OD$_{600}$ = 0.5, and 2 μl culture was injected into soft agar plates (½ BHI with 0.3% agar) in technical duplicate with a minimum of three biological replicates. The swimming diameter was measured after 24 h at 30˚C for *B. subtilis*, or every 24 h for five days at 37˚C for *C. difficile*, and replicate values were averaged. A two-tailed Student's *t*-test was performed to determine statistical significance, comparing wild-type and mutant motilities, or ANOVA with Dunnett's multiple comparisons test for multiple comparisons (GraphPad Prism v9.0).

## Animal studies

Male and female Syrian golden hamsters (*Mesocricetus auratus*; 6–8 weeks old) were purchased from Charles River Laboratories and housed in sterile, individual cages in an animal biosafety level 2 facility. Seven days prior to challenge with *C. difficile* spores, hamsters were treated with one dose of clindamycin (30 mg kg$^{-1}$ body weight) by oral gavage to facilitate susceptibility to *C. difficile* infection. Prior to infection, spores were heated for 20 min at 55˚C and allowed to cool to room temperature before inoculation. Hamsters were inoculated with 5,000 spores of strains 630Δ*erm* or the *spo0E* mutant, prepared as previously described and stored in 1X PBS 0.1% BSA solution.[34,46] Negative control animals were given clindamycin to induce susceptibility to disease but were not inoculated with *C. difficile* spores. Animal experiments were performed with two independent spore preps in two separate cohorts.

Animals were monitored regularly for progression of disease symptoms (lethargy, weight loss, wet tail, diarrhea). After administration of spores, fecal samples were collected daily to determine total *C. difficile* CFU, and an additional fecal sample from each hamster was collected 24 h after infection for *in vivo* toxin assays. Hamsters were considered moribund if they had lost 15% of their highest weight, or presented advanced symptoms of lethargy, wet tail, or diarrhea. Hamsters that met these criteria were euthanized in accordance with the American Veterinary Medical Association guidelines. At the time of death, cecal contents were collected for toxin assays and enumeration. *C. difficile* in both fecal and cecal contents were enumerated by plating samples on TCCFA agar.[47,48] Recovered CFU from cecal and fecal contents of animals infected with 630Δ*erm* or the *spo0E* mutant were assessed by a Student's two-tailed *t* test, and differences in hamster survival time between 630Δ*erm* or *spo0E* infection were analyzed by log-rank test in GraphPad Prism v9.0.

## Co-immunoprecipitation

*C. difficile* cultures of 630Δ*erm* expressing the vector control (MC324), *spo0E*-3xFLAG (MC1968), or *spo0A*-3xFLAG (MC1003) were grown on 70:30 sporulation agar supplemented with 2 μg ml$^{-1}$ thiamphenicol as described above. After 12 hours of growth, cells were harvested from plates, pelleted, washed with 1X PBS, and stored at -80˚C. Cells were then thawed on ice and resuspended in mBS/THES buffer (50 mM HEPES, 25 mM CaCl$_2$, 250 mM KCl, 50 mM Tris-HCl pH 7.5, 2.5 mM EDTA, 140 mM NaCl, 0.7% Protease Inhibitor Cocktail II [Sigma], 0.1% Phosphatase Inhibitor Cocktail II [Sigma], and 1% glycerol) supplemented with DNase I (Sigma) and RNase A (Thermo-Fisher). Cells were lysed by 25 freeze-thaw cycles consisting of repetitive 3 min incubations in a dry ice-ethanol bath followed by 2 min in a 37˚C water bath. Cell debris were pelleted at 21,300 x *g* at 4˚C, and lysates were collected. Equilibrated anti-FLAG beads (Sigma) were washed in TBS buffer and then subsequently washed in mBS/THES buffer. Sample lysates were then incubated with washed anti-FLAG beads on a mechanical rotor for 4 h at room temperature. Beads were then collected in a 1.5 mL Protein LoBind tube (Eppendorf), and lysates were saved for analysis. Beads were washed three times in mBS/THES buffer, transferred to a new 1.5 mL Protein LoBind tube, then washed three times with 1X PBS, finally suspended in 1X PBS, and stored at -20˚C.

## Silver staining and western blotting

To visualize total protein or recombinant FLAG-tagged Spo0E and Spo0A in protein pulldown samples, silver staining or western blotting, respectively, was performed on wildtype, Spo0E-3xFLAG, and Spo0A-3xFLAG samples collected during co-immunoprecipitation. Briefly, lysates or eluates were suspended in 1X sample buffer (10% glycerol, 62.5 mM Upper Tris, 3% SDS, 5 mM PMSF, and 5% 2-mercaptoethanol), separated by SDS-PAGE using 4–20% TGX precast gels (BioRad). Silver staining was performed using the Pierce Silver Staining Kit according to manufacturer's instructions (Thermo-Fisher).

For western blotting following co-immunoprecipitation, proteins were recovered from anti-FLAG M2 magnetic beads (Sigma) with a brief incubation in 100 μg/ml 3XFLAG peptide (Sigma). Lysates (input) and eluates were mixed with Laemmli Buffer (BioRad) and separated on SDS-PAGE using 4–20% TGX precast gels (BioRad). Proteins were transferred to a 0.2 μm nitrocellulose membrane, and Spo0E-3XFLAG was detected with anti-FLAG M2 antibody (Sigma), Spo0A was detected with anti-FLAG [30], and RstA-HA was detected with anti-HA (Sigma). Silver stained gels and western blots were imaged using a BioRad ChemiDoc MP System.

## On-bead digestion for LC-MS/MS

On-bead digestion in preparation for LC-MS/MS was performed following an established protocol.[49] Digestion buffer (50 mM NH$_4$HCO$_3$) was added to the beads, and the mixture was treated with 1 mM dithiothreitol (DTT) at room temperature for 30 minutes, followed by 5 mM iodoacetimide (IAA) at room temperature for 30 minutes in the dark. Proteins were then digested overnight with 2 μg of lysyl endopeptidase (Wako) at room temperature and further digested overnight with 2 μg trypsin (Promega) at room temperature. Resulting peptides were desalted with HLB column (Waters) and were dried under vacuum.

## LC-MS/MS

The data acquisition by LC-MS/MS was adapted from a published procedure [50]. Derived peptides were resuspended in 0.1% trifluoroacetic acid (TFA) and were separated on a Water's

Charged Surface Hybrid (CSH) column (150 μm internal diameter (ID) x 15 cm; particle size: 1.7 μm). The samples were run on an EVOSEP liquid chromatography system using the 15 samples per day preset gradient (88 min) and were monitored on a Q-Exactive Plus Hybrid Quadrupole-Orbitrap Mass Spectrometer (Thermo Fisher). The mass spectrometer cycle was programmed to collect one full MS scan followed by 20 data dependent MS/MS scans. The MS scans (400–1600 m/z range, $3 \times 10^6$ AGC target, 100 ms maximum ion time) were collected at a resolution of 70,000 at m/z 200 in profile mode. The HCD MS/MS spectra (1.6 m/z isolation width, 28% collision energy, $1 \times 10^5$ AGC target, 100 ms maximum ion time) were acquired at a resolution of 17,500 at m/z 200. Dynamic exclusion was set to exclude previously sequenced precursor ions for 30 seconds. Precursor ions with +1, and +7, and +8 or higher charge states were excluded from sequencing.

## MaxQuant

Label-free quantification analysis of protein pulldown samples was adapted from a published procedure.[50] Spectra were searched using the search engine Andromeda, integrated into MaxQuant, against *C.difficile* Uniprot database (3,969 target sequences). Methionine oxidation (+15.9949 Da), asparagine and glutamine deamidation (+0.9840 Da), and protein N-terminal acetylation (+42.0106 Da) were variable modifications (up to five per peptide); cysteine was assigned as a fixed carbamidomethyl modification (+57.0215 Da). Only fully tryptic peptides were considered with up to two missed cleavages in the database search. A precursor mass tolerance of ±20 ppm was applied prior to mass accuracy calibration and ±4.5 ppm after internal MaxQuant calibration. Other search settings included a maximum peptide mass of 6,000 Da, a minimum peptide length of 6 residues, 0.05 Da tolerance for orbitrap and 0.6 Da tolerance for ion trap MS/MS scans. The false discovery rate (FDR) for peptide spectral matches, proteins, and site decoy fraction were all set to 1%. Quantification settings were as follows: re-quantify with a second peak finding attempt after protein identification has completed; match MS1 peaks between runs; and a 0.7 min retention time match window was used after an alignment function was found with a 20-minute RT search space. Quantitation of proteins was performed using summed peptide intensities given by MaxQuant. The quantitation method only considered razor plus unique peptides for protein level quantitation.

## LC-MS/MS data analysis

To determine statistical significance between experimental (Spo0A-FLAG, Spo0E-FLAG) and negative control groups, Perseus software (Version 1.6.15.0) was used to analyze Intensity data.[51] Intensity values were $\log_2$ transformed, and data was filtered to remove: contaminants, proteins only identified by site, and reverse hits. Imputation of data was performed based on normal distribution with downshift of 1.8 and width of 0.3. A two-way Student's *t*-test was performed to determine significantly enriched proteins between the experimental group (Spo0A-3xFLAG or Spo0E-3xFLAG) and negative control. P-values were then adjusted with permutation based false discovery rate (FDR) for proteins that were identified in at least three of four replicates. Scatter plots were generated in Perseus. Proteins enriched with a P-value $\leq 0.05$ were considered statistically significant. Proteins were additionally filtered by a cutoff of 1.2 $\log_2$ transformed Intensity ratio relative to the negative control.

## Phylogenetic comparisons

Putative Spo0E orthologs were identified using PSI-BLAST to probe for the conserved Spo0E SxxxD motif, and AlphaFold to search for predicted Spo0E-like proteins.[22,23] Protein alignments were performed using ClustalW under default settings.[52] An unrooted Neighbor-

Joining tree using full-length Spo0E and Spo0E-like protein sequences was created using MEGA11.[53] Predicted 3D protein structures were generated using Phyre2, and the resultant output PDB files were edited using PyMOL (The PyMOL Molecular Graphics System, Version 2.4.0 Schrödinger, LLC).[54] Protein accession numbers of Spo0E-like proteins used in the phylogenetic analysis are as follows: *C. difficile* (WP_009891746.1), *Intestinibacter bartlettii* (WP_216572026.1), *Paeniclostridium sordellii* (WP_021126610.1), *Bacillus subtilis* (NP_389247.1), *Streptococcus pneumoniae* (CJR48991.1), *Staphylococcus epidermidis* (WP_145378230.1), *Clostridium botulinum* (WP_106898918.1), *Clostridium perfringens* (UBL05073.1), *Pseudomonas amygdali* (WP_016766164.1), *Escherichia coli* (WP_224654603.1), *Vibrio vulnificus* (TDL93146.1), *Mycobacterium tuberculosis* (WP_079178562.1), *Methanosaeta* (OPY55450), *Chlamydia trachomatis* (CRH64375.1), *Bacillus anthracis* (PFB78764.1), *Bacillus cereus* (AUZ26151.1), *Listeria monocytogenes* (ECO1678074.1), *Mycobacteroides abscessus* (SLB39125.1), and *Rhodococcus qingshengii* (SLB39125.1).

## Protein interaction modeling

AlphaFold models were generated from sequences retrieved from the Uniprot database through the AlphaFold-multimer ColabFold with default parameters (*C. difficile* Spo0E, A0A7Y0LUW8, Spo0A (aa 5–115), Uniprot ID: Q18B74, and RstA, UniParc ID: UPI0003FBD6A6; *B. subtilis* Spo0E, Uniprot ID: P05043 and Spo0A, Uniprot ID: P06534 [55]. Figures were generated in PyMol (PyMOL Molecular Graphics System, Version 2.0 Schrödinger, LLC.). The bonds between proteins were determined through the PyMol "polar contacts" and were determined to be confident through the UCSF ChimeraX program, developed by the Resource for Biocomputing Visualization and Informatics at the University of California, San Francisco (https://www.cgl.ucsf.edu/chimerax/), via the predicted aligned error (PAE) values between two given residues.[56] The PAE values represent the Angstrom distance ranging from 0 to 30 Angstroms where a value less than 5 is considered confident. The AlphaFold Multimer resource is available through an open source Colab code. ().

## Supporting information

**S1 Table. Bacterial Strains and plasmids.**
(DOCX)

**S2 Table. Filtered proteins identified in Spo0E-FLAG co-immunoprecipitation.**
(DOCX)

**S3 Table. Filtered proteins identified in Spo0A-FLAG co-immunoprecipitation.**
(DOCX)

**S4 Table. *C.difficile* projected Spo0E interactions with RstA and Spo0A are mutually exclusive.**
(DOCX)

**S5 Table. *C. difficile* and *B. subtilis* respective Spo0E-Spo0A projected interactions.**
(DOCX)

**S6 Table. Predicted *C. difficile* Spo0A and *B. subtilis* Spo0E interactions.**
(DOCX)

**S7 Table. Predicted *B. subtilis* Spo0A and *C. difficile* Spo0E residue interactions.**
(DOCX)

**S8 Table. Oligonucleotides.**
(DOCX)

**S9 Table. Vector and strain construction.**
(DOCX)

**S1 Fig. Construction and confirmation of the *spo0E* mutant.**
(PDF)

**S2 Fig. Spo0E and Spo0A co-purify with regulators of sporulation, toxin, and motility.**
(PDF)

**S3 Fig. Predicted *C. difficile* Spo0E interactions with Spo0A and RstA.**
(PDF)

**S4 Fig. Spo0E of *C. difficile* and *B. subtilis* do not cross-complement *spo0E* mutant motility or sporulation phenotypes.**
(PDF)

**S5 Fig. Spo0E-like proteins are conserved and prevalent in Gram-positive and Gram-negative bacteria.**
(PDF)

## Acknowledgments

We give special thanks to members of the McBride Lab and to Charlie Moran for suggestions during the completion of this work and preparation of this manuscript and to K.L. Nawrocki for help in the construction of pMC228.

## Author Contributions

**Conceptualization:** Michael A. DiCandia, Adrianne N. Edwards, Shonna M. McBride.

**Data curation:** Michael A. DiCandia, Ysabella B. Alcaraz, Cheyenne D. Lee, Shonna M. McBride.

**Formal analysis:** Michael A. DiCandia, Adrianne N. Edwards, Ysabella B. Alcaraz, Cheyenne D. Lee, Pritha Bagchi, Shonna M. McBride.

**Funding acquisition:** Shonna M. McBride.

**Investigation:** Michael A. DiCandia, Adrianne N. Edwards, Marcos P. Monteiro, Cheyenne D. Lee, Germán Vargas Cuebas, Shonna M. McBride.

**Methodology:** Adrianne N. Edwards, Pritha Bagchi.

**Project administration:** Shonna M. McBride.

**Supervision:** Adrianne N. Edwards, Shonna M. McBride.

**Validation:** Adrianne N. Edwards, Shonna M. McBride.

**Visualization:** Michael A. DiCandia, Adrianne N. Edwards, Ysabella B. Alcaraz, Marcos P. Monteiro, Cheyenne D. Lee.

**Writing – original draft:** Michael A. DiCandia, Shonna M. McBride.

**Writing – review & editing:** Michael A. DiCandia, Adrianne N. Edwards, Ysabella B. Alcaraz, Marcos P. Monteiro, Cheyenne D. Lee, Germán Vargas Cuebas, Shonna M. McBride.

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
