## [Decision Letter · Decision Letter 0]

27 Nov 2023

Dear Dr McBride,

Thank you very much for submitting your manuscript "A Conserved Switch Controls Virulence, Sporulation, and Motility in C. difficile" for consideration at PLOS Pathogens. As with all papers reviewed by the journal, your manuscript was reviewed by members of the editorial board and by independent reviewers. In light of the reviews (below this email), we would like to invite the resubmission of a revised version that takes into account the reviewers' comments.

The Reviewers were enthusiastic about the finding that Spo0E regulates multiple processes in both C. difficile and B. subtilis. The finding that Spo0E interacts directly or indirectly with Spo0A and RstA in C. difficile is an important contribution, but Reviewers 1 and 3 request additional verification of these mass spectrometry-based co-IP experiments either in the form of validating the AlphaFold-predicted interactions or through another experimental method. They raised additional points regarding alternative interpretations of the data that can be readily addressed through textual changes. If these items are addressed, I hope to make a final decision without needing to send the manuscript out for review again.

Sincerely,

Aimee Shen

Guest Editor

PLOS Pathogens

Michael Wessels

Section Editor

PLOS Pathogens

Kasturi Haldar

Editor-in-Chief

PLOS Pathogens

orcid.org/0000-0001-5065-158X

Michael Malim

Editor-in-Chief

PLOS Pathogens

orcid.org/0000-0002-7699-2064

The Reviewers were enthusiastic about the finding that Spo0E regulates multiple processes in both C. difficile and B. subtilis. The finding that Spo0E interacts directly or indirectly with Spo0A and RstA in C. difficile is an important contribution, but Reviewers 1 and 3 request additional verification of these mass spectrometry-based co-IP experiments either in the form of validating the AlphaFold-predicted interactions or through another experimental method. They raised additional points regarding alternative interpretations of the data that can be readily addressed through textual changes. If these items are addressed, I hope to make a final decision without needing to send the manuscript out for review again.

Reviewer's Responses to Questions

**Part I - Summary**

Reviewer #1: This manuscript describes new phenotypical observations and mechanistic explanations of how the small protein, Spo0E, impacts sporulation, motility, and toxin production in C. difficile. TargeTron disruption of spo0E was generated, validated by sequencing, and complemented to demonstrate that sporulation frequency and TcdA+TcdB toxin is negatively impacted by Spo0E under lab conditions, and it has a modest negative impact on motility. The mutant displays an enhanced virulence phenotype in an animal model of infection. To identify proteins that interact with Spo0E that could explain phenotypes beyond sporulation, Co-IP was conducted. RstA, a transcription factor, was a top candidate. AlphaFold modeling was used to predict interactions between Spo0E, RstA and Spo0A. Residues known to confer interactions between Spo0E and Spo0A from prior studies in Bacillus subtilis were identified in the model’s protein-protein interface, providing confidence in the Spo0E-Spo0A prediction. The role of Spo0E in B. subtilis motility was investigated, as it had not been questioned before, and indeed found to impact this characteristic. However, cross-complementation studies (placing C.diff spo0E into B.sub, and vice versa) found that they could not substitute for one another. Yet, spo0E-like proteins are identified among a variety of bacteria, based on sequence and structural predictions, and therefore indicate these small proteins are likely to be important regulators.

The primary weakness of this study is seen in its reliance on Alphafold predictions between Spo0E and RstA. No evidence is provided to support the notion that Spo0E and RstA actually interact. Nevertheless, the manuscript is easy to follow, the included experiments are technically sound, and the findings are highly significant.

Reviewer #2: The manuscript by DiCandia et al. examines the phosphatase Spo0E in Clostridium difficile. In B. subtilis, Spo0E represses the function of Spo0A, the master regulator of entry into sporulation, but the function of Spo0E in C. difficile is not known. The authors report that in C. difficile, spo0E also represses sporulation but unexpectedly discovered that Spo0E also increases motility and toxin production. Interestingly, an effect of Spo0E on motility (albeit, the opposite effect, but still Spo0A-independent) was conserved and previously overlooked in B. subtilis. Most strikingly, the authors report that Spo0E-like proteins are broadly conserved in bacteria, suggesting that a similar regulatory mechanism exists in other, non-sporulating, cells.

Reviewer #3: Overall this is a well written manuscript describing the important role of Spo0E in regulating sporulation, virulence, toxin production and motility. This is mostly a descriptive study showing the importance of Spo0E in regulating these phenotypes. The authors use Co-IP mass spec to identify interacting partners of Spo0E and Spo0A. There is not significant follow-up demonstrating the importance of these interactions.

**Part II – Major Issues: Key Experiments Required for Acceptance**

Reviewer #1: Major Comments:

1. Foremost, the AlphaFold prediction needs to be validated experimentally by some method, preferably showing the predicted interface between Spo0E and RstA is true.

2. Line 169; I disagree with the statement, “the number of spo0E mutant bacteria present in cecum was similar to the WT strain.” It seems there are obviously less bacteria in the distribution of mutant-infected hamsters, compared to the wildtype infected animals. This is interesting given the enhanced virulence of the mutant, which is likely due to elevated toxin production.

3. It appears from figure 3 that Spo0E interacts with a monomer of RstA. Is it known whether RstA needs to form a dimer to bind DNA, like most RRNPP transcriptional regulators with a DNA-binding domain? Would the Spo0E-RstA interaction be possible if RstA were in a dimeric complex? If RstA were to require dimer formation to bind DNA (likely) might the mechanism of Spo0E work by preventing RstA dimer formation? Is there a way to easily test whether Spo0E impacts RstA dimerization?

4. As Spo0E was found to interact with both Spo0A and RstA by Co-IP, and because RstA is reported to impact sporulation by an unknown mechanism, a model is put forth (line 206, Fig. S7) that Spo0E serves as an intermediate between the two regulatory proteins. It could be argued that Spo0E doesn’t need to physically bridge RstA to Spo0A for RstA to have an impact on sporulation. Perhaps Spo0E interacts with one partner at a time and the effect on sporulation is merely an outcome of whether Spo0E is free to engage Spo0A. I suggest including this alternative scenario that Spo0E may not need to function as intermediate between RstA and Spo0A.

Reviewer #2: In general, this was a well-written paper with largely well-designed experiments that should be easily accessible for a broad audience. I have only some relatively cosmetic suggestions for the authors.

Major comments:

1. Fig. 2B. Statistical analysis suggests that the p value is greater than 0.05 (therefore ‘not significant’), and the authors conclude that increase in toxin production was not due to increased colonization or carriage. That said, the difference in the spread of the data between the two groups is striking and it really looks like deleting spo0E results in reduced number of bacteria in the cecum., depending on the particular animal. Is it possible that increased toxin production *causes* a decreased C. difficile burden in a fraction of animals? In other words, what is the virulence/pathogenesis consequence of increased toxin production? Not requesting additional experiments here, just proposing a different interpretation of the results that the authors may or may not wish to address.

2. Fig. 2, Table 1. If modeling predicts that Spo0E, RstA, and Spo0A form a complex, any idea why RstA did not copurify with Spo0A (was RstA present, but below the cutoff value in the LC-MS/MS data for the Spo0A co-IP)?

3. Consider moving the model figure in Fig. S7 to the main text as a new Fig. 5.

Reviewer #3: 1. Some insight into what the important interactions of Spo0E would strengthen the paper Ideally this would be point mutants that disrupt interaction with Spo0A or RtsA specifically but may be outside the scope. A less difficult experiment would be to construct a spo0E rstA double mutant to determine if it resembles a spo0E or rtsA phenotype. This could provide some insight to which interactions are more important.

2. The CO-IP mass spec experiments should be confirmed using another assay.

**Part III – Minor Issues: Editorial and Data Presentation Modifications**

Reviewer #1: Minor comments.

Could the authors comment on why the motility phenotype is so modest as compared to toxin production if both are thought to be mediated by RstA and SigD?

Line 203. ‘Complement’ probably isn’t the best word choice, maybe state as the converse experiment?

Reviewer #2: Minor comments:

1. Line 106. Please define “RRNPP” upon first use.

2. Fig. 1B, 1D, Fig. 4. Consider eschewing the use of bar graphs and instead use scatter graphs (similar to Fig. 2B-D) so that the reader may appreciate the spread of data.

3. The complementation data in Fig. S2 is very important. Consider moving Fig. S2A-B into Fig. 1.

Reviewer #3: 1. I would like to see the complementation data in the main text.

2. Do Spo0E homologs from other organisms complement either B. subtilis or C. difficile?

3. It would be helpful to know if the flag-tagged proteins are functional (do they complement mutants).

PLOS authors have the option to publish the peer review history of their article (what does this mean?). If published, this will include your full peer review and any attached files.

Reviewer #1: No

Reviewer #2: No

Reviewer #3: No
---

## [Decision Letter · Decision Letter 1]

16 Apr 2024

Dear Dr McBride,

Thank you very much for submitting your manuscript "A Conserved Switch Controls Virulence, Sporulation, and Motility in C. difficile" for consideration at PLOS Pathogens. As with all papers reviewed by the journal, your manuscript was reviewed by members of the editorial board and by independent reviewers. The reviewers appreciated the attention to an important topic. Based on the reviews, we are likely to accept this manuscript for publication, providing that you modify the manuscript according to the review recommendations.

The newly added co-immunoprecipitation data greatly strengthened this interesting manuscript. The Reviewers and I are in agreement that the article should be accepted, but there is a request for very slight changes in a revised manuscript that can be easily addressed. It would be nice to include the silver-stained gel of the pull-downs into the supplemental (as Reviewer 3 notes, the methods described silver staining). Reviewer 3 also had questions about the band labeled as "RstA dimer" in the co-IP. For example, there is a band that may run at a similar size in the Spo0A western blot (but it is hard to tell with the MW markers shown).

In addition, to avoid confusion of readers who may be quickly glancing at the figures to indicate that the structures shown are AlphaFold models in the figure legends and figure legend title. Please change the title of Figure 5 so that it reads "C. difficile Spo0E is predicted to interact with functional domains of Spo0A and RstA." Similarly, for supplemental figure 3, please revise the legend title to read "Predicted C. difficile Spo0E interactions with Spo0A and RstA."

Sincerely,

Aimee Shen

Guest Editor

PLOS Pathogens

Michael Wessels

Section Editor

PLOS Pathogens

Michael Malim

Editor-in-Chief

PLOS Pathogens

orcid.org/0000-0002-7699-2064

The newly added co-immunoprecipitation data greatly strengthened this interesting manuscript. The Reviewers and I are in agreement that the article should be accepted, but there is a request for very slight changes in a revised manuscript that can be easily addressed. It would be nice to include the silver-stained gel of the pull-downs into the supplemental (as Reviewer 3 notes, the methods described silver staining). Reviewer 3 also had questions about the band labeled as "RstA dimer" in the co-IP. For example, there is a band that may run at a similar size in the Spo0A western blot (but it is hard to tell with the MW markers shown).

In addition, to avoid confusion of readers who may be quickly glancing at the figures to indicate that the structures shown are AlphaFold models in the figure legends and figure legend title. Please change the title of Figure 5 so that it reads "C. difficile Spo0E is predicted to interact with functional domains of Spo0A and RstA." Similarly, for supplemental figure 3, please revised the legend title to read "Predicted C. difficile Spo0E interactions with Spo0A and RstA."

Reviewer Comments (if any, and for reference):

Reviewer's Responses to Questions

**Part I - Summary**

Reviewer #1: (No Response)

Reviewer #3: In the initial review I raised two concerns. The first was a confirmation of the interaction between SpoOE and RstA which was provided and convincing. They were unable to confirm with genetics that this interaction is biologically relevant. Otherwise they have addresssed the majority of the concerns well.

**Part II – Major Issues: Key Experiments Required for Acceptance**

Reviewer #1: (No Response)

Reviewer #3: (No Response)

**Part III – Minor Issues: Editorial and Data Presentation Modifications**

Reviewer #1: (No Response)

Reviewer #3: I am not convinced based on what the authors have written that what is labeled as RtsA dimer is really a dimer. Is it possible that this has not entered the gel (there are not standards labeled above the "dimer)? If looked at closely there are "dimer" bands present in the Eluate of all 3 samples. It would be beneficial to include the silver stain in the supplemental figures as it is mentioned in the methods but I did not see this in the main or supplemental figures.

PLOS authors have the option to publish the peer review history of their article (what does this mean?). If published, this will include your full peer review and any attached files.

Reviewer #1: No

Reviewer #3: No

Figure Files:

Data Requirements:

Reproducibility:

References:

---

## [Editor Report · Decision Letter 2]

25 Apr 2024

Dear Dr McBride,

We are pleased to inform you that your manuscript 'A Conserved Switch Controls Virulence, Sporulation, and Motility in C. difficile' has been provisionally accepted for publication in PLOS Pathogens.

Best regards,

Aimee Shen

Guest Editor

PLOS Pathogens

Michael Wessels

Section Editor

PLOS Pathogens

Michael Malim

Editor-in-Chief

PLOS Pathogens

orcid.org/0000-0002-7699-2064

Thank you for your revised submission. The newly added silver stained gel is a fantastic addition to the paper. We greatly appreciate your attention to the Reviewers' comments and thank you for submitting this excellent study.
---

## [Editor Report · Acceptance letter]

4 May 2024

Dear Dr McBride,

We are delighted to inform you that your manuscript, "A Conserved Switch Controls Virulence, Sporulation, and Motility in C. difficile ," has been formally accepted for publication in PLOS Pathogens.

Best regards,

Michael Malim

Editor-in-Chief

PLOS Pathogens

orcid.org/0000-0002-7699-2064